# Preliminary Mechanical Analysis of Rubber-Cement Composites Suitable for Additive Process Construction

**Matteo Sambucci** [1,2], **Danilo Marini** [2], **Abbas Sibai** [1] and **Marco Valente** [1,2,*]

1   Department of Chemical and Material Engineering, Sapienza University of Rome, 00184 Rome, Italy;
    matteo.sambucci@uniroma1.it (M.S.); abbas.sibai@uniroma1.it (A.S.)
2   INSTM Reference Laboratory for Engineering of Surface Treatments, Department of Chemical and Material
    Engineering, Sapienza University of Rome, 00184 Rome, Italy; danilo.marini@uniroma1.it
*   Correspondence: marco.valente@uniroma1.it; Tel.: +39-06-4458-5582

**Abstract:** Additive manufacturing for cementitious materials represents the most attractive frontier in the modern context of Construction 4.0. In addition to the technological progress of printing systems, the development of functional and low environmental impact printable mixtures is one of the current challenges of digital fabrication in building and architectural fields. This paper proposes a preliminary physical-mechanical analysis on environmentally friendly mortars, compatible with the extrusion-based printing process, made up of recycling rubber aggregates deriving from end-of-life tires. In this study, two groups of rubber particle samples (0–1 mm rubber powder and 2–4 mm rubber granules) were used to partially/totally replace the mineral fraction of the reference printable mixture. Four tire rubber powder-granules proportions were investigated and control mortar (100% sand) was also prepared to compare its properties with those of the rubber-cement samples in terms of printability properties, mechanical strength, ductility, and structural isotropy. Based on the experimental results, the rubber aggregates increase the mixture fluidity, promoting better inter-layer adhesion than the neat mix. This leads to greater mechanical isotropy. As already investigated in other research works on Rubber-Concrete technology, the addition of rubber particles increases the ductility of the material but reduces its mechanical strength. However, by correctly balancing the fine and coarse rubber fraction, promising physical-mechanical performances were demonstrated.

**Keywords:** additive manufacturing; tire rubber-cement compounds; tire recycling; printability; mechanical anisotropy; physical-mechanical characterization

## 1. Introduction

Digitization and automation of production processes are key aspects of the fourth industrial revolution, defined as Industry 4.0. Nowadays, the integration of advanced technologies with traditional manufacturing methods is a strategy deeply rooted in various industrial sectors such as biomedicine, automotive, aerospace, and design. In this framework, the construction industry is facing big challenges characterized by the adoption of digital technologies, sensor systems, advanced manufacturing apparatuses, and smart building materials. This transformation, which by analogy to the manufacturing sector has been called Construction 4.0, will enable construction companies to improve productivity and design flexibility, reduce project delays and cost overruns, manage complexity, and enhance safety, quality, and resource-efficiency [1]. Building Information Modelling (BIM) is one of such innovative processes, which is based on the integration of engineering, economic, and managerial tools, in the design and fabrication phases of construction work. With BIM technology, an accurate virtual model of a building known as a Building Information Model is digitally constructed

and used to support the design, material procurement, manufacturing, maintenance, and facility management of the building after completion [2]. In addition to BIM, Additive Manufacturing (AM) has recently gained popularity in the construction sector. Unlike traditional methods of casting cementitious material into molds or formworks, AM combines digital technology and new insights from material science to allow freeform construction without the use of expensive formwork [3]. According to the definition by Buswell et al., freeform constructions are: "Processes for integrated building components which demonstrate added value, functionality, and capabilities over and above traditional methods of construction" [4]. This strategy has several manufacturing implications: (a) automation; (b) "design before build" and geometric freedom; (c) higher degrees of structural optimization; (d) development of new building materials and processes. Besides, there are also attractive benefits that AM aims to bring to the building-architectural industry [5,6]:

- Reduction in injury rates by eliminating dangerous activities.
- Increasing sustainability in construction: less material waste and reduced $CO_2$ emissions.
- Creation of high-end technology-based jobs.
- Increasing architectural freedom which would enable more sophisticated designs for engineering and aesthetic purposes.
- Enhance multi-functionality for structural/architectural elements. Thanks to this feature, construction components can be designed to tune their properties according to the type of application to which they are addressed (weight reduction, improvement of mechanical properties, acoustic damping, thermal insulation).

Research activities on AM for the construction industry led to the development of two main digital fabrication techniques: powder-based and extrusion-based printing processes. In the powder-based process, a print head deposits an inorganic or organic liquid binder on a loose powder bed to bond the part layer by layer. This method has the potential to make robust and durable building components at a reasonable speed to satisfy this industrial demand and with very complex geometries (resolutions on the order of mm) [3,7]. D-Shape technique [8] and Stone Spray project [9] are some examples of powder-based AM. However, some process criticalities hinder their maximum potential for application in the construction industry [3,6,7]: (a) limited range of cement-based materials suitable for powder-based 3D printers; (b) difficulty in introducing structural reinforcements; (c) high sensitivity of cementitious powders to humidity, limiting the recycling of unbound material; (d) need to perform several post-manufacturing operations (such as infiltration of binder solution or additional curing steps) that can adversely affect the production rate. On the other hand, the extrusion-based printing process has been researched extensively by several academic teams and companies over the last decade [10–14]. In this technique, a digitally controlled nozzle, mounted on a gantry or robotic arm, extrudes fresh cementitious mix layer by layer until the complete creation of the pre-designed model. Rheology and components balance of the printable compounds are crucial to achieving a high-quality printing process. The material must be extrudable and able to maintain its shape once deposited over the printing bed. Besides, the deposited layers should not collapse under the load of subsequent layers and a good inter-layer adhesion must be ensured for better-hardened properties. The growing interest in this approach is justified by some examples of small-scale and large-scale building applications designed and developed by some scientific projects (Table 1).

Currently, the challenges of digital construction are focused on three aspects [6]:

- Technological optimization of the printing process (automated implementation of structural reinforcements or integration of multi-deposition devices);
- Rigorous studies on the development of building units with functional morphologies;
- Study and design of novel printable materials with low environmental impact (use of eco-friendly binders or industrial wastes as mixture aggregates).

**Table 1.** An overview of the main industrial and academic research projects operating in cement materials extrusion-based additive manufacturing (AM).

| Research Team or Company | Project | Printed Product | Source |
|---|---|---|---|
| Eindhoven University of Technology | 3DCP | 3D printed pedestrian bridge elements | [10] |
| XTREE | 3D Pavilion | Urban furniture with functional properties | [11] |
| WASP | GAIA House | 30 m$^2$ module with improved energy efficiency | [12] |
| Singapore Centre for 3D Printing | Saddle and Dome-shaped surfaces | 3D printed curved concrete panels | [13] |
| TU Dresden, Institute of Construction Materials | CONprint3D | On-site monolithic construction | [14] |

The latter is the key topic of this research work. This paper presents a study based on the development of cement mortars, suitable for AM, modified with rubber aggregates deriving from end-of-life tires (ELTs).

The European Tyre Recycling Association (ETRA) Draft Model [15] states that since the beginning of the 21st century, 3.2 million/year of ELTs have accumulated in Europe. In the recycling sector, ongoing research is aimed to develop clean and innovative solutions for waste tires disposal. Among the most promising alternative is the incorporation of ground tire rubber (GTR), deriving from the mechanical grinding of ELTs, in cementitious matrices. This strategy, commonly known as Rubber-Concrete (RC) technology, is widely studied in the traditional building materials field, and is aiming to produce a new class of more sustainable cement-based compounds with specific engineering properties. Many studies highlight that the hydrophobic nature and viscoelastic properties of GTR particles improve the thermal and acoustic insulation peculiarities of the material, increase its ability to absorb plastic energy under loading, reduce the permeable porosity, and enhance the structural lightweight [16–19]. However, the main weak point of the mixtures concerns the remarkable reduction in mechanical strength, mainly related to the chemical-physical incompatibility between rubber and cement paste. For this reason, research activities on RC technology are widespread to identify building-architectural applications suitable for the physical-mechanical requirements provided by these rubberized materials.

The possibility to combine RC technology with AM processes is the innovative aspect of this work. The main benefit that can be drawn is related to the possibility of exploiting the architectural freedom of digital construction to design and develop building components with highly functional complex shapes and geometries, then, minimize the mechanical weakness of the material by enhancing its performances through the topological optimization approach. The present manuscript summarizes preliminary experimental results of 3D printable tire rubber-cement mortars obtained by partial and total volume replacement of the mineral aggregate with two GTR types, in 0–4 mm grain size range: rubber powder (RP) and rubber granules (RG). The influence of polymer aggregates on morphological features, printing properties, and physical-mechanical behavior of the compounds was investigated, with emphasis on the effects of particle size and volume content.

## 2. Materials and Methods

### 2.1. Material Properties and Mix Proportions

Constituent materials for control cementitious mixes (no GTR aggregates) included a Type I Portland cement, limestone sand with a 0.4 mm maximum size, 18 m$^2$/g fineness Silica fume-based thixotropic additive (TA), Polycarboxylate ether-based superplasticizer (SA), 0–0.18 mm size aliphatic-based water reducing agent (WRA), and Calcium oxide-based expansive agents (EA).

Rubber aggregates, provided by ETRA, are produced by ambient mechanical grinding of waste vehicle tires into small particles. Two types of not pretreated GTR fractions were used: rubber powder (Figure 1a) and rubber granules (Figure 1b).

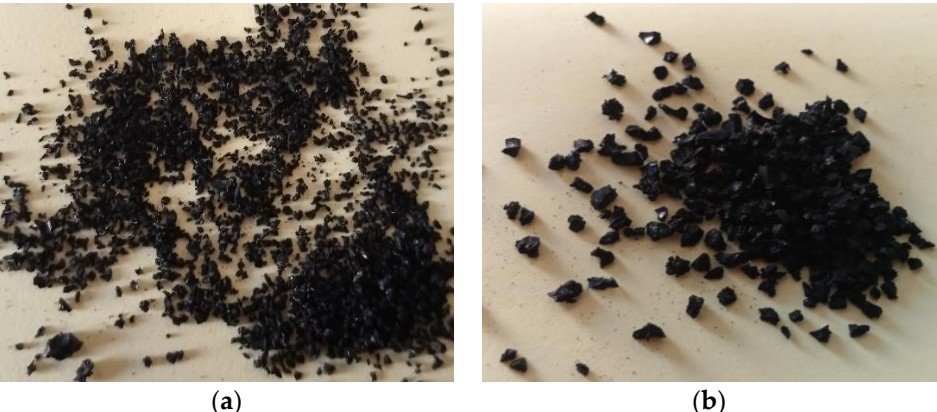

**Figure 1.** Ground tire rubber (GTR) aggregates used in this research: rubber powder (**a**) and rubber granules (**b**).

The specific gravity and the particle size distribution of "as-received" rubber aggregates are shown in Table 2.

**Table 2.** Properties of ground tire rubber (GTR) particles.

| GTR | Specific Gravity (kg/m$^3$) | Particle Size Distribution (mm) |
|---|---|---|
| Rubber powder (RP) | 1209 | 0–1 |
| Rubber granules (RG) | 1195 | 2–4 |

The control mix was used as the basis for preparing four rubberized compounds specified by S50-P50, P100, P50-G50, and P25-G75. The sand-rubber volume replacements in each rubber-cement mix are listed below:

- S50-P50: 50% by volume of sand—50% by volume of RP.
- P100: 100% by volume of RP.
- P50-G50: 50% by volume of RP—50% by volume of RG.
- P25-G75: 25% by volume of RP—75% by volume of RG.

Except for the S50-P50 mix, all the investigated rubber-based mixtures were based on the total replacement of the mineral aggregates with the polymer ones. Mixture proportioning specifications are detailed in Table 3.

**Table 3.** Cement-based mixes design.

| Mix | Cement (kg/m$^3$) | Water (kg/m$^3$) | Sand (kg/m$^3$) | RP (kg/m$^3$) | RG (kg/m$^3$) | TA + SA + WRA + EA (kg/m$^3$) |
|---|---|---|---|---|---|---|
| Control | 800 | 300 | 1100 | 0 | 0 | 152 |
| S50-P50 | 800 | 280 | 550 | 150 | 0 | 152 |
| P100 | 800 | 260 | 0 | 300 | 0 | 152 |
| P50-G50 | 800 | 250 | 0 | 150 | 160 | 152 |
| P25-G75 | 800 | 230 | 0 | 75 | 240 | 152 |

As for the water content, because of the hydrophobic nature of rubber particles and larger size than limestone sand used in this work, the water requirement of rubberized compounds was lower

compared to the control mix. To ensure proper rheology for the extrusion-based deposition process, the water amount, shown in Table 3, was optimized and selected following the printability tests described below.

3D Printability Tests and Samples Manufacturing

Printability tests were conducted at the Dept. of Materials, Environmental Sciences, and Urban Planning, Marche Polytechnic University (Ancona, Italy), using a COMAU 3-axis robotic arm (Figure 2) equipped with a circular extrusion nozzle (Ø = 10 mm). More technical details on the extrusion system are reported in [18,19].

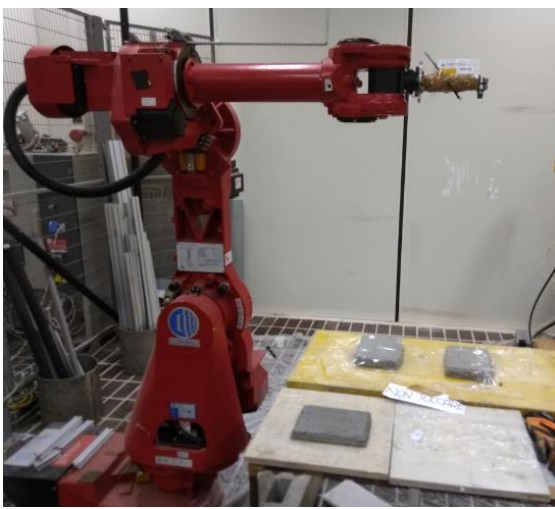

**Figure 2.** Three-axis robotic arm-based printing device.

The experimentation aimed to evaluate, through a trial-and-error approach, the optimum water dosage to consider the rubber-cement mixtures as 3D printable. The proper rheology of the fresh mixes was selected evaluating three fundamental printing process indicators, in agreement with the technical recommendations reported in Papachristoforou et al.'s research work [20]:

- Extrudability. This can be defined as the ability of the material to be deposited regularly and without interruptions/blockages in the extrusion nozzle.
- Buildability (or shape stability). This is the material's ability to retain its shape, as per the extruder dimension, under pressure from the upper layers without the occurrence of collapse phenomena. Five printed layers of cementitious material are the "target-condition" for defining the mix as printable.
- Inter-layer adhesion. This is related to the bond strength between the printed layers. High inter-layer adhesion implies a compact material without voids or defects between the deposited filaments. This requirement is closely related to the mechanical properties of the printed samples. The structural homogeneity of the object promotes its isotropic behavior in terms of mechanical strength.

For this purpose, both the control and rubberized mortars were 3D printed to produce 6-layer parallelepiped structures (220 mm × 160 mm × 55 mm) with a linear extrusion speed of 33 mm/s. The size and geometry of the printed samples were selected to verify the buildability requirement mentioned above and to ensure the adequate extraction of the specimens for physical-mechanical characterization. The pump flow rate was adjusted accordingly to ensure a smooth deposition. Subsequently, from each hardened 3D printed slab (Figure 3), two sets of samples were extracted by sawing in a diamond cutter: 40 mm-side cubes and (six specimens for each mix) and 40 mm × 40 mm × 220 mm beams (four specimens for each mix). Cubes were used for bulk density measurements and mechanical compressive tests. Beams were tested under four-point flexural tests.

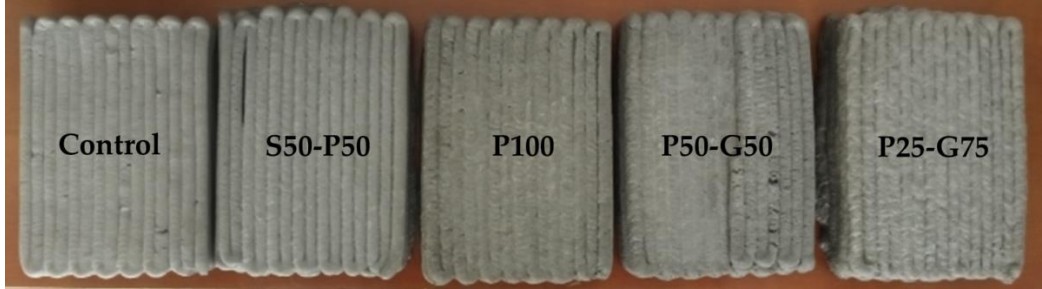

**Figure 3.** Hardened 3D printed slabs resulting from printability tests.

The flowchart in Figure 4 summarizes the experimental procedure implemented for the manufacture of 3D printable rubber-cement mixtures and the preparation of the samples.

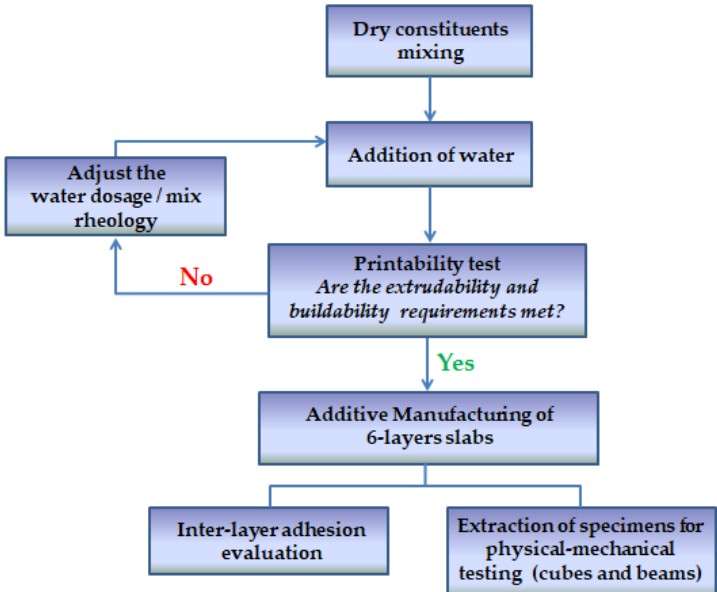

**Figure 4.** Manufacturing and printability testing of rubber-cement mixtures: experimental procedure.

## 2.2. Experimental Testing

### 2.2.1. Microscopical Analysis on GTR and Microstructure Characterization

To obtain morphological information about the GTR aggregates used in rubber-cement mixes, microscopic observations were undertaken using a Leica MS5 stereomicroscope (Leica Camera AG, Wetzlar, Germany) and a Nikon Eclipse L150 optical microscope (Nikon Corporation, Tokyo, Japan) (Figure 5a) at 5× and 6.3× magnifications, respectively. In all techniques above, rubber particles were fixed on a metal support through an adhesive tape (Figure 5b) and analyzed to identify their morphology and surface texture. The microphotographs were acquired using Lucia measurement imaging software.

Field Emission Gun electron microscopy (FEG-SEM) was used to investigate the cement mortars microstructure and the rubber-cement interface properties. Due to non-conductive properties, small fragment samples were polished and coated with a conductive carbon film using a Leica EM SCD005 vacuum sputter coater to avoid charging phenomena and image instability during micrograph collection. FEG-SEM analyses were performed using a TESCAN MIRA3 FEG-SEM (Tescan Company, Brno, Czech Republic) in Secondary Electron (SE) acquisition mode.

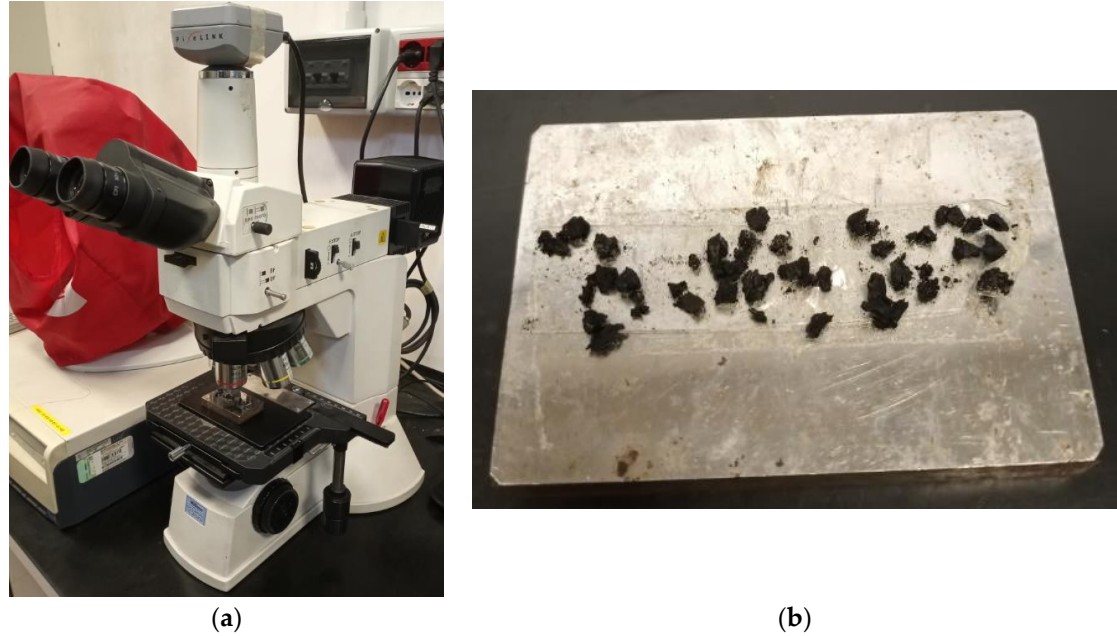

(**a**)      (**b**)

**Figure 5.** Microscopical analysis of GTR particles (**a**) and sample preparation (**b**).

### 2.2.2. Bulk Density Evaluation

Bulk density was determined as the ratio between the oven-dry mass of cubic samples (110 °C for 48 h) and their geometric volume in accordance with BS 1881-114 standard [21]. For each investigated mix, bulk density results were computed from the average of four specimens.

### 2.2.3. Mechanical Characterization: Compressive Test

Compression strengths and stress–strain response of the control and rubberized compound were investigated according to the ASTM C109/C109M-20a standard test method [22], using a Zwick-Roell Z150 machine (150 kN capacity, Zwick Roell Group, Ulm, Germany) at a loading rate 1 mm/min and a 20 N pre-load. To evaluate the effect of inter-layer adhesion and structural compaction on the mechanical isotropy, cubic samples were loaded in two different directions (X-loading and Z-loading directions), as shown in Figure 6, and the compressive strength average values were noted considering three specimens in each loading condition.

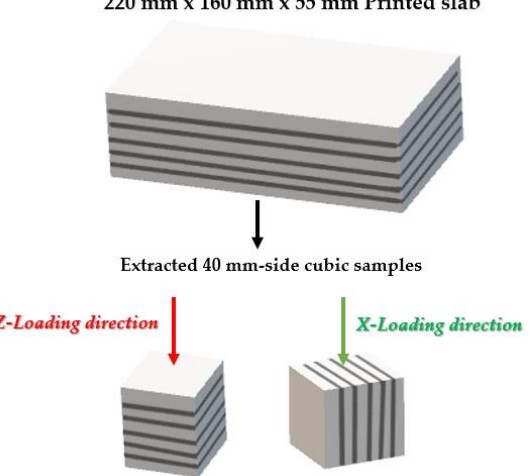

**Figure 6.** Testing direction for the compressive test.

### 2.2.4. Mechanical Characterization: Four-Point Flexural Test

Flexural strength and Young's modulus of both the control and rubber-cement mortars were measured using the four-point flexural test according to the ASTM C348 standard test method [23]. Beams (four samples for each mix) were tested by a Zwick-Roell Z010 machine with 10 kN capacity, using a support span of 180 mm at a loading rate of 1 mm/min. Unlike the compression tests, all cement samples were tested under a single condition (Z-Loading direction), applying a bending load perpendicular to the deposition plane.

## 3. Results and Discussion

### 3.1. 3D Printability Test: A Qualitative Analysis

As mentioned earlier, the rheology of cement mortars is crucial for the AM process, since the print quality as well as the hardened properties are directly related to their fresh properties. The mix formulations shown in Table 3 are the result of 3D printability tests performed by gradually varying the water dosage until obtaining optimal matching in terms of extrudability, buildability (Figure 7a), and inter-layer adhesion. AM of the slabs took place through a regular deposition process, without interruptions or structural failures. The grain size and morphological irregularity of the GTR particles promote greater surface roughness of the samples than the control sample, but this effect does not affect the proper printability properties of the material. To evaluate the hardened samples compaction and the presence of inter-layer defects, the internal section of the slabs (Figure 7b) was visually observed.

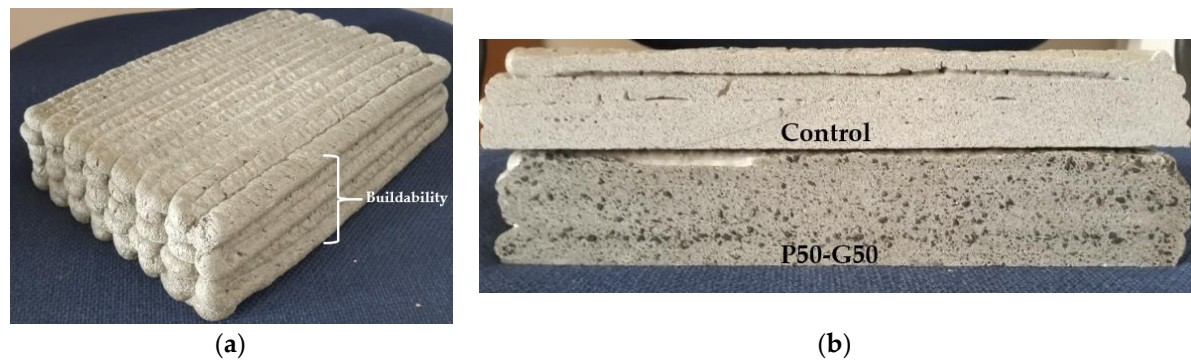

         (**a**)                                  (**b**)

**Figure 7.** Proper buildability in the P50-G50 sample (**a**) and visual inspection of inter-layer adhesion in the control and P50-G50 mixes (**b**).

As shown in Figure 7b, the comparison between the control and the rubberized samples (in this case, the P50-G50 mix) shows an evident difference in structural compaction. The formation of inter-layer voids between the unrubberized filaments (control) stems from the higher surface tension of the extruded mixture. Conversely, the non-polar properties of GTR aggregates decrease the overall surface tension of the compounds (due to low rubber-cement interfacial interactions), resulting in an easier flow that allows filling of the voids before hardening. This evidence is in good agreement with those reported by previous studies [18,24], where the incorporation of rubber particles into cement matrices increases the fresh material fluidity.

### 3.2. Microscopical Analysis on GTR and Microstructure Characterization

Figure 8a,b show the stereomicroscope images of RP and RG, respectively.

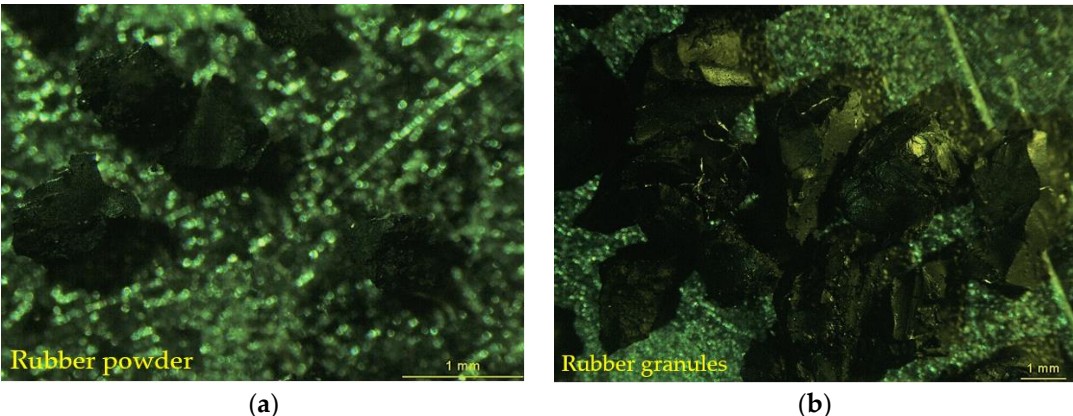

**Figure 8.** Stereomicrographs of GTR aggregates used in this research: rubber powder (RP) (**a**) and rubber granules (RG) (**b**).

As well highlighted in previous research [18,25,26], the size of GTR particles and the physical-mechanical properties of the rubber-cement composite are strictly interdependent. Depending on the tire rubber mechanical grinding degree, the aggregates exhibit a specific surface texture which affects the adhesion with the cement matrix and their tendency to trap air during the mixing process of the fresh compound. The rough and angular surface texture of the fine GTR fraction (<3 mm size) promotes an efficient anchoring mechanism with cement paste, due to the penetration of the cementitious binder into the particles' structural "jaggies" (Figure 9a). However, the high specific surface improves their tendency to incorporate air, promoting material density reduction and, consequently, a remarkable loss in mechanical strength performance. On the other hand, the aspect ratio in coarse GTR aggregates (Figure 9b) is less than the fine ones, implying a weaker cohesion with the cement matrix and a lower tendency to adsorb gas. Compared to the finest fraction, coarse GTR particles functionality is attributable to the greater ability to hinder the cracks propagation, reducing the stress concentrations in the matrix and improving the material deformability [27].

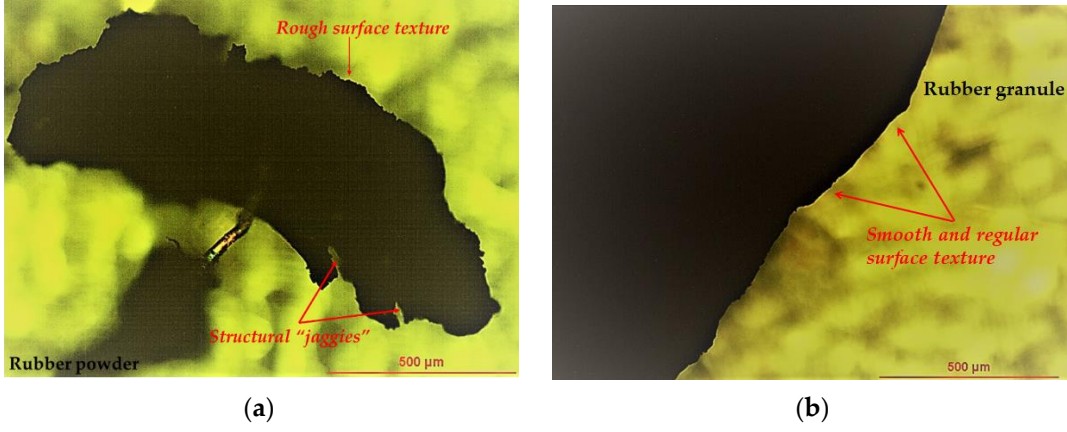

**Figure 9.** Surface texture of GTR aggregates: rubber powder (**a**) and rubber granule (**b**).

The FEG-SEM micrographs in Figure 10 show the different interface bonding in the case of RP cement (Figure 10a) and RG cement (Figure 10b) adhesions.

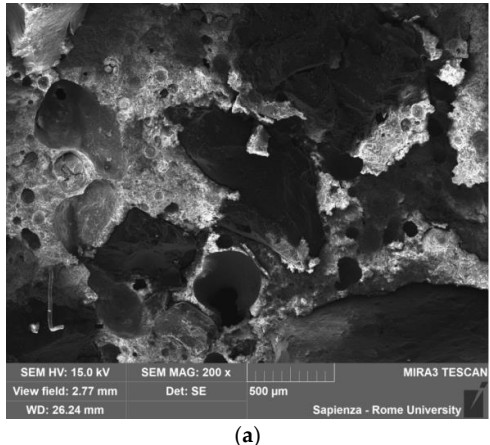

(**a**)

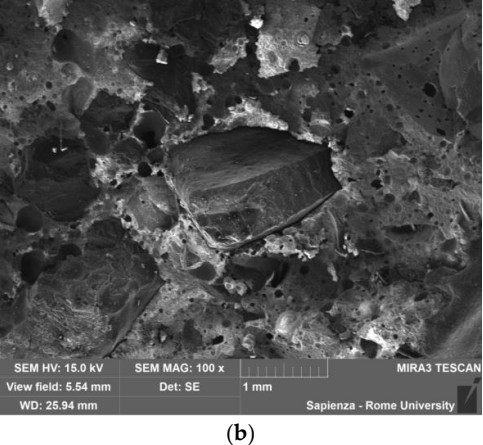

(**b**)

**Figure 10.** FEG-SEM micrographs on GTR–cement interface adhesion: rubber powder (**a**) and rubber granule (**b**).

According to the previous discussion, it is possible to observe a greater interfacial cohesion between the RP and the cement matrix compared to the RG cement interface. This confirms the strong relationship between particle morphology and adhesion properties.

*3.3. Bulk Density Evaluation*

Bulk density experimental results are reported in Figure 11

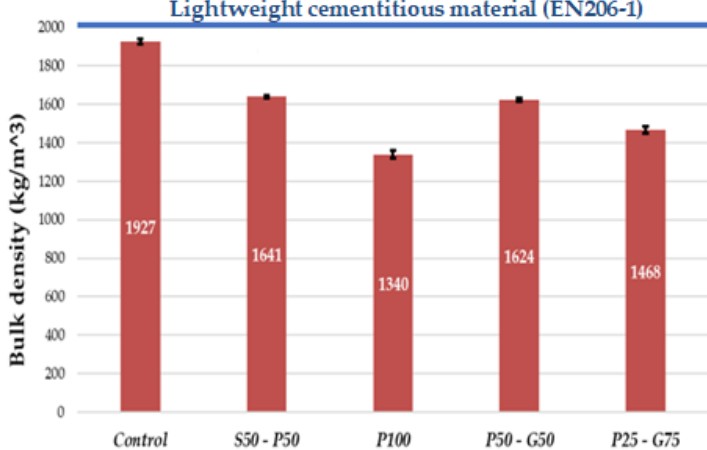

**Figure 11.** Bulk density of the all printable cement mortars.

The bulk density of printable mortars varied from 1340 to 1927 kg/m$^3$. It can be noticed that this range complies with the European specifications for lightweight concrete materials (oven-dry density range of approximately 300 to a maximum of 2000 kg/m$^3$) [28]. Due to low thermal conductivity and unit weight, the lightweight building material can be used to prove high insulation properties and to reduce the dead load of a structure [29].

The non-polar nature of GTR aggregates results in the ability to repel water and entrap air on the surface of the particles, which would subsequently increase the air content and thus, decrease the material density. The density reduction in rubberized compounds is also related to the lower specific weight of the polymer aggregates compared to the mineral ones, therefore, a higher replacement level implies a more relevant unit weight lowering [30]. In the P100 sample, the most noticeable reduction in bulk density occurs (−30%). As mentioned in Section 3.2, the high specific surface of RP promotes air incorporation, improving the structural lightweight of the material.

### 3.4. Mechanical Characterization: Compressive Test

Directional mechanical strength is one of the inherent properties of the extrusion-based AM process due to layer-wise manufacturing [31]. In this regard, compressive strength was recorded in two loading conditions (see Figure 6) to determine the effect of structural compaction on the mechanical anisotropy of the samples. Furthermore, experimental testing allowed the evaluation of the variation of mechanical strength as a function of the type and amount of GTR aggregate incorporated in the mixtures.

Figure 12 presents the average compressive strength results in X-loading and Z-loading directions for each mix.

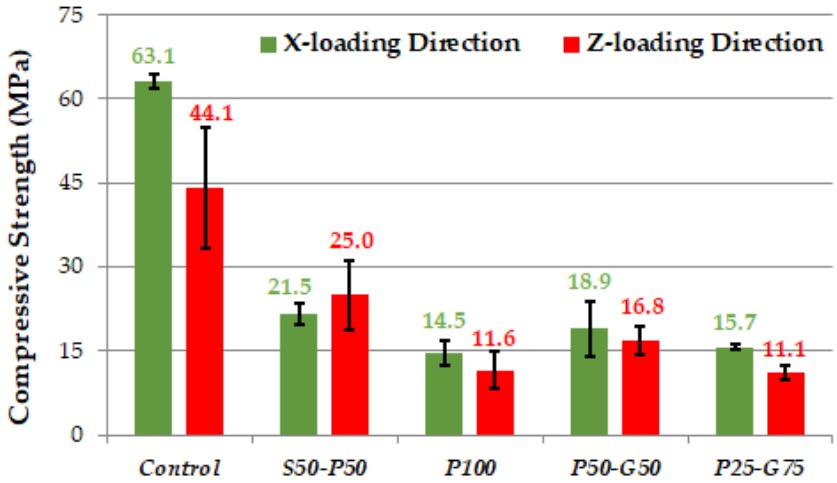

**Figure 12.** Compressive strength of both the control and rubber-cement samples tested in two loading directions.

Testing printed cubes in two loading directions showed significant compressive strength divergence (~30%) in the control sample. In this case, mechanical anisotropy is strictly attributable to the morphological-structural properties of the hardened mixture. As noted in Section 3.1, the presence of inter-layer defects in the control slab emphasizes the "layering" effect, promoting a heterogeneous transfer and distribution of the applied load. In GTR-modified cement mixes, a very similar mechanical behavior is observed, due to the good compaction of the samples. Then, the orientation dependence of strength properties is less marked. Except for the S50-P50 mix, comparing the mechanical behavior of the samples under the two loading conditions, a higher average compressive strength is observed in the case of compression load applied along the filaments (X-loading direction) than Z-loading direction (compression load perpendicular to the printed layers). According to the results from Panda et al. [32],

loading applied in a plane perpendicular to layers allows direct transfer and distribution of stress uniformly throughout the cross-section.

Regardless of direction loading, the inclusion of GTR particles significantly decreases the compressive strength compared to the neat sample. Weak rubber-cement interfacial adhesion and the remarkable reduction in material bulk density, related to the increase in internal porosity, are the main reasons for this mechanical behavior [17,26,27,30]. The dependence between density reduction and mechanical strength worsening in RC mixtures is well demonstrated in the empirical relationship (Equation (1)) proposed by Benazzouk et al. [33]:

$$\sigma_c = k \times e^{(0.0031 \times \rho)} \tag{1}$$

In this equation, $\sigma_c$ represents the static compressive strength (MPa), $\rho$ is the material density (kg/m$^3$), and $k$ is an empirical parameter related to the type of cement matrix.

The lowest mechanical strength reduction occurs in the S50-P50 and P50-G50 mixes. The mineral aggregates, partially incorporated in the S50-P50 mix, acts as a structural reinforcement more effectively than the polymer particles. The P50-G50 sample exhibits interesting mechanical behavior. The greater strength compared to the other totally modified rubberized samples could be attributable to the efficient synergy between the GTR fractions: RP may have a filler effect to increase the compactness of the rubber-cement compound, while RG hinder the excessive formation of internal porosity and minimize micro-cracks coalescence, improving the material ductility. Figure 13 shows the failure pattern of the control, P100, and P50-G50 samples.

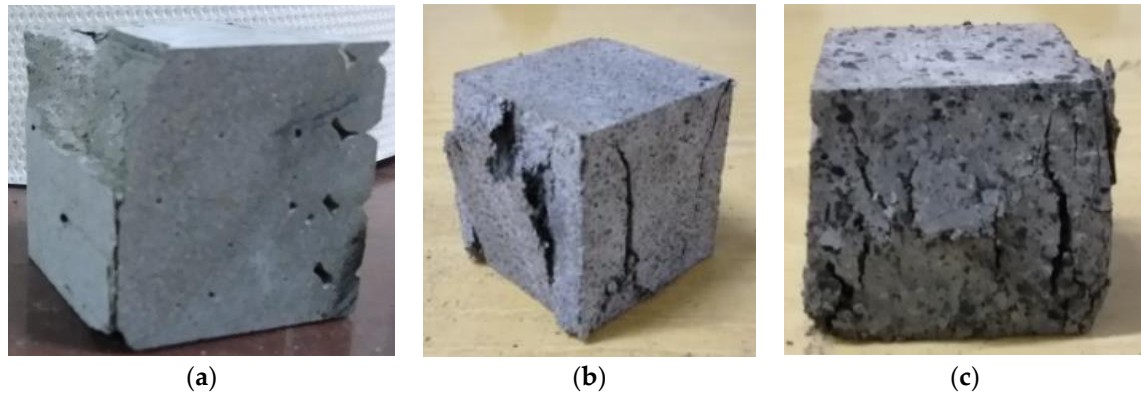

| (**a**) | (**b**) | (**c**) |

**Figure 13.** Failure pattern after compressive test: Control (**a**), P100 (**b**), and P50-G50 (**c**).

Rubber-cement specimens present large lateral deformations compared to the plain sample. During the unloading process, the elastomeric behavior of tire particles decreases the internal friction among the cement elements and recovers extra strain [34]. The ductile character is accentuated in the RG-based compound (P50-G50), where coarse GTR aggregates provide more attenuated cracks propagation.

*3.5. Mechanical Characterization: Four-Point Flexural Test*

Results of the four-point flexural test are given in Figure 14. The flexural strength reduction follows the same trend as compressive strength. The most important factor in reducing flexural performance, as well as the compressive ones, is the poor compatibility between rubber particles and cement paste. The decrease in Young's modulus when GTR aggregates are incorporated can be justified by the fact that the elastic modulus of the two-component (aggregate and cement matrix) composite depends on Young's modulus of the aggregates and their volumetric proportion. Therefore, the lower stiffness of rubber particles than sand will result in overall elastic modulus reduction, directly related to the volume of rubber added [35]. This explains the identical elasto-mechanical behavior in GTR-cement mortars based on the same rubber-sand replacement level (P100, P50-G50, and P25-G75). Ductility is a

very desirable structural requirement because it allows load redistribution and enables the cement material to have the capacity to deform and support mechanical stress when cracking occurs.

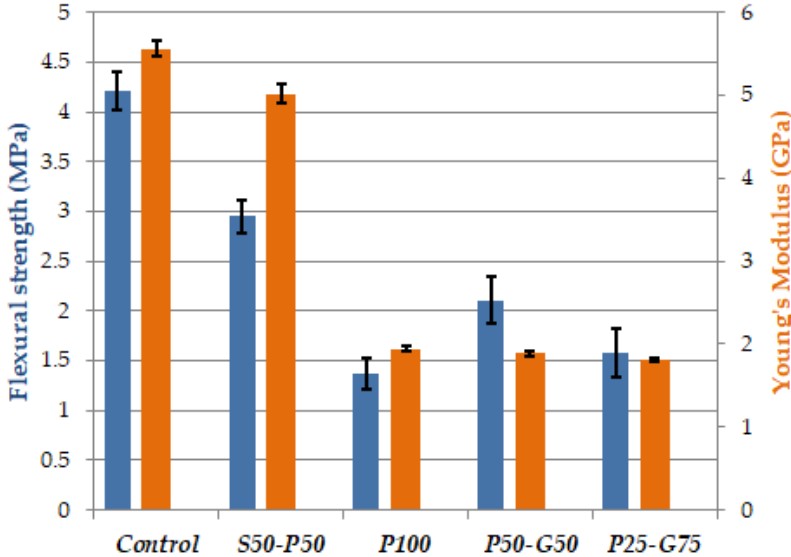

**Figure 14.** Four-point flexural strength results: flexural strength and Young's modulus.

## 4. Conclusions

This scientific contribution reports a preliminary analysis on eco-friendly cement mortars, suitable for extrusion-based AM, modified with recycled tire rubber aggregates. The possibility of applying RC technology to digital fabrication processes is the innovative value of this research. Based on the experimental investigation, the following conclusions can be drawn.

- The replacement of mineral aggregates by GTR particles does not alter the proper printability of cement-based mixtures. Polymer aggregates modify the surface tension of the fresh compound, increasing its fluidity and ensuring a more efficient layer-by-layer deposition process in terms of adhesion between the filaments, compaction of the hardened material, and mechanical isotropy.
- The incorporation of lightweight and hydrophobic rubber particles implies a reduction in the unit weight of the cement mortars. However, by properly combining the two polymer fractions used in this work (RP and RG), it is possible to obtain a less relevant density loss with consequent advantages in mechanical strength performances.
- Low density and poor rubber-cement interfacial bonding are the main causes of the mechanical strength loss in rubber-cement composites. However, by exploiting the synergy between fine (RP) and coarse fractions (RG), it was possible to observe better strength performances (P50-G50 mix). Partial sand-rubber replacement (S50-P50 mix) provides good mechanical requirements but could represent a less efficient strategy in terms of enhancing the tire recycling process and saving natural resources.
- More ductile behavior is observed for rubberized mortars compared to plain cement specimens under compressive and flexural testing. These can represent interesting requirements in applications where structural deformability is a primary requirement (flexible paving bricks, flexible sub-base for pavements, anti-shock barriers).

Future activities will be devoted to three research field:

- Complete the experimental campaign on 3D printable rubber-cement mixtures, evaluating the effect of GTR fractions on the acoustic and thermal properties. The authors have already published a piece of research concerning the durability performance of the materials investigated in this work [18].

- Study and analyze potential chemical-physical treatments to improve rubber-cement adhesion properties. Compatibilizing treatments and surface modification methods are widely used in composites materials technology, where matrix reinforcement adhesion is a fundamental requirement to improve the physical-mechanical properties of the material [36,37].
- Explore the frontiers of AM and exploit its design flexibility to enhance the use of RC technology in the construction and architectural fields.

**Author Contributions:** Conceptualization M.V. and M.S.; Investigation M.S. and D.M. and A.S.; Formal analysis M.S., D.M.; Original Draft Preparation: M.S.; writing—review and editing M.S., D.M., M.V.; Supervision M.V.; All authors have read and agreed to the published version of the manuscript.

**Funding:** This research was also performed thanks to "Sapienza" University direct financing, for PhD student Matteo Sambucci, called "Avvio alla Ricerca".

**Acknowledgments:** The authors would like to acknowledge the scientific support by Valeria Corinaldesi and Glauco Merlonetti (Marche Polytechnic University).

**Conflicts of Interest:** The authors declare no conflict of interest.

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
