# Peer review of "Preliminary Mechanical Analysis of Rubber-Cement Composites Suitable for Additive Process Construction"

_jcs, doi:10.3390/jcs4030120_

Round 1
Reviewer 1 Report
Manuscript entitled “Preliminary mechanical analysis of rubber-cement composites suitable
for additive process construction”, authored by Matteo Sambucci, Danilo Marini, Abbas Sibai, Marco Valente has been carefully read and analysed in context of publication in the journal.
In my opinion the manuscript present high level of the expertise in the field of composite materials based on cement-polymer systems. I suggest to accept the manuscript in the present form.
Author Response
Cover letter for Reviewer 1
Manuscript entitled “Preliminary mechanical analysis of rubber-cement composites suitable for additive process construction”, authored by Matteo Sambucci, Danilo Marini, Abbas Sibai, Marco Valente has been carefully read and analysed in context of publication in the journal. In my opinion the manuscript present high level of the expertise in the field of composite materials based on cement-polymer systems. I suggest to accept the manuscript in the present form.
We are pleased to thank the reviewer for his positive opinion on the manuscript and for his valuable feedback.
Reviewer 2 Report
This work is an attempt to investigate the mechanical properties of rubber-cement composites that used for constriction purposes. It is an interesting topic however the following points can improve the work
- In Line 177, the authors mentioned that they used parallelepiped structures for the development of materials. the question is why they selected this structure? As you may know, the internal structure of materials has a significant effect on mechanical performance.
- This is good if the authors can add a flowchart as a summary of the methodology. It helps the readers to find out the general outline of the paper
- The conclusion part is like a summary, not a conclusion. Please revise it.
Author Response
Cover letter for Reviewer 2
In Line 177, the authors mentioned that they used parallelepiped structures for the development of materials. the question is why they selected this structure? As you may know, the internal structure of materials has a significant effect on mechanical performance.
The parallelepiped structure was selected to verify the buildability requirement and to ensure an adequate extraction of the samples for the material characterization. These details have been added in the manuscript (see Lines 173 and 181)
This is good if the authors can add a flowchart as a summary of the methodology. It helps the readers to find out the general outline of the paper
As suggested by the reviewer, a summary flowchart on the experimental methodology implemented in the work has been added (see Figure 4). Thanks for the valuable suggestion.
The conclusion part is like a summary, not a conclusion. Please revise it.
“Conclusion” section has been edited and revised in accordance with the reviewer’s comment.
The authors have carefully reviewed and corrected the English language and style.